# Physicochemical Properties, Antioxidant Markers, and Meat Quality as Affected by Heat Stress: A Review

**DOI:** 10.3390/molecules28083332

**Published:** 2023-04-10

**Authors:** Bochra Bejaoui, Chaima Sdiri, Ikram Ben Souf, Imen Belhadj Slimen, Manel Ben Larbi, Sidrine Koumba, Patrick Martin, Naceur M’Hamdi

**Affiliations:** 1Laboratory of Useful Materials, National Institute of Research and Pysico-Chemical Analysis (INRAP), Technopark of Sidi Thabet, Ariana 2020, Tunisia; 2Department of Chemistry, Faculty of Sciences of Bizerte, University of Carthage, Zarzouna, Bizerte 7021, Tunisia; 3Research Laboratory of Ecosystems & Aquatic Resources, National Agronomic Institute of Tunisia, Carthage University, 43 Avenue Charles Nicolle, Tunis 1082, Tunisia; 4Department of Animal Sciences, National Agronomic Institute of Tunisia, Carthage University, 43 Avenue Charles Nicolle, Tunis 1082, Tunisia; 5Laboratory of Materials, Molecules, and Application, Preparatory Institute for Scientific and Technical Studies, B.P. 51, La Marsa, Tunis 2078, Tunisia; 6LR13AGR02, Higher School of Agriculture, University of Carthage, Mateur 7030, Tunisia; 7Unité Transformations & Agroressources, ULR7519, Université d’Artois-UniLaSalle, F-62408 Bethune, France

**Keywords:** antioxidant status, biochemical properties, chemical composition, heat stress, meat quality

## Abstract

Heat stress is one of the most stressful events in livestock life, negatively impacting animal health, productivity, and product quality. Moreover, the negative impact of heat stress on animal product quality has recently attracted increasing public awareness and concern. The purpose of this review is to discuss the effects of heat stress on the quality and the physicochemical component of meat in ruminants, pigs, rabbits, and poultry. Based on PRISMA guidelines, research articles were identified, screened, and summarized based on inclusion criteria for heat stress on meat safety and quality. Data were obtained from the Web of Science. Many studies reported the increased incidences of heat stress on animal welfare and meat quality. Although heat stress impacts can be variable depending on the severity and duration, the exposure of animals to heat stress (HS) can affect meat quality. Recent studies have shown that HS not only causes physiological and metabolic disturbances in living animals but also alters the rate and extent of glycolysis in postmortem muscles, resulting in changes in pH values that affect carcasses and meat. It has been shown to have a plausible effect on quality and antioxidant activity. Acute heat stress just before slaughter stimulates muscle glycogenolysis and can result in pale, tender, and exudative (PSE) meat characterized by low water-holding capacity (WHC). The enzymatic antioxidants such as superoxide dismutase (SOD), catalase (CAT), and glutathione peroxidase (GPx) act by scavenging both intracellular and extracellular superoxide radicals and preventing the lipid peroxidation of the plasma membrane. Therefore, understanding and controlling environmental conditions is crucial to successful animal production and product safety. The objective of this review was to investigate the effects of HS on meat quality and antioxidant status.

## 1. Introduction

Food quality is a very complex and broad concept that has changed rapidly in recent years. Recently, it has represented the set of all food properties that are closely related to physicochemical properties, texture, and taste that are acceptable and satisfying to consumers. Previous studies showed that consumers’ concerns about food quality have been increasing, in recent decades, particularly concerning the perceived healthiness of food [1,2,3]. Meat and various meat products are important sources of high-quality proteins, fats, and minerals as essential nutrients [4,5]. As animal welfare concerns grow and consumers become more conscious of food quality, there is an increasing focus on improving the quality, safety, and nutritional value of meat. Based on current knowledge, increasing global warming and climate change appear to pose a potential threat to food security in the coming decades [6,7]. Heat stress (HS) is one of the major impacts of climate change (CC) on livestock raised in both intensive and extensive production systems [8]. This is especially true given its negative impact on behavior, immune response, gut integrity, productivity, and meat quality [9]. Heat stress is a form of hyperthermia (elevated body temperature) in which the body’s physiological systems are unable to regulate body temperature within normal limits [8]. It is well known that, at temperatures higher than an animal’s thermoneutral zone, heat stress can affect animal welfare, performance, and product quality. In the same context, Rana et al. [10] reported that CC, particularly HS, could impact meat safety as well as organoleptic quality. Early studies have shown that acute or chronic HS can deteriorate meat quality. This is due to decreased protein synthesis, increased fat deposition, and increased ante-/postmortem glycolytic metabolism associated with the production of reactive oxygen species (ROS) [11,12,13]. In the case of broilers, the exposure of chickens to high temperatures can induce a lower ultimate pH with variation in meat color, water-holding capacity (WHC), and tenderness of meat [11,13,14,15], resulting in lower consumer acceptability. In addition to flavor and oxidative stability, the safety of meat is also a very important issue (foodborne illness of microbial origin) [16]. Previously, Lowe et al. [17], Kadim et al. [18], and Macías et al. [19] showed that the increased incidence of HS in animals has been reported in many countries, especially warmer parts of the world, compromising animal welfare, and having flow-on effects on meat quality traits. Although the effects of HS may vary in severity and duration, the exposure of animals to elevated temperatures increases the concentration of omega-6 fatty acids in muscle [6,11]. However, the effects of HS on meat quality across animal species and breeds are inconsistent and are likely due to differences in heat resistance [4,20]. These impacts on carcass characteristics have been reported in many species, and HS has been well known to reduce feed intake and carcass yield in poultry [13,21], pigs [22], goats [23], cattle [24], and lambs [25]. Furthermore, the causes of variation in meat quality characteristics between different animal species, muscle fibers within animals, and even meat parts in response to stress are not yet fully understood [26]. Recent studies have shown that the magnitude of carcass trait changes can vary between species, but overall, reduced carcass yields due to HS result in significant economic losses to the livestock industry [20,26,27]. Long-term HS also results in decreased subcutaneous fat and lower intramuscular fat (IMF) levels, which may assist the animal with better heat dissipation rates [28]. However, a small number of reviews have been conducted to depict the effects of HS on meat safety and quality.

In this context, this review will focus on the effects of heat stress on the quality, and the physicochemical component of meat.

## 2. Heat Stress and Carcass Characteristics

Carcass and meat quality depend on both intrinsic and extrinsic factors. The intrinsic factors include species, breed, gender, age, and slaughter weight. The extrinsic factors include weaning, diet, and stress [29,30]. This latter can be attributed to the breeding conditions, the transport, or the environment. To maintain euthermia, heat-stressed animals activate some physiologic and metabolic adjustments at the expense of growth, reproductive, and productive aspects [31,32]. For this purpose, homeothermic animals reduce their feed intake to lower metabolic heat production [33]. However, such an adaptive response has implications on carcass characteristics in monogastric and ruminants as well. Carcass yield, carcass fat deposition, and intramuscular fat content were reported to be decreased in poultry [13,21], pigs [22,34], sheep [35], goats [23], and cattle [24,35]. A reduction in subcutaneous fat was also reported in pigs subjected to chronic HS, to enhance heat dissipation [36]. Holinger et al. [37] described that the carcass of heat-stressed pigs had reduced lean meat percentage and thicker backfat. The extent of changes indeed depends on the species, but it is known that the economic losses caused by carcass yield loss are great [38]. At the same time as fat content dropped, acetyl coenzyme A carboxylase enzyme, L (+) P-hydroxy acyl CoA dehydrogenase, and lipolytic enzyme activities were decreased [22,36]. Pearce et al. [22] demonstrated that this decrease is independent of heat stress-induced reductions in feed intake. Nevertheless, no significant effects on intramuscular fat were reported in broilers subjected to heat stress for 3 weeks [39] and heat-stressed goats for 1 month [30]. Similarly, Mader et al. [40] and Ponnampalam et al. [41] did not report any significant difference between chronic heat stress (1 week) and thermal neutral groups in subcutaneous fat in cattle and carcass fat scores in lambs. These findings may be attributed to the ability of the breeds to cope with heat stress conditions, and to the duration and severity of HS [30]. However, it is noteworthy to mention that although the carcass weight was significantly affected by heat stress-induced feed intake reduction, the impacts on carcass composition are confusing and need further investigation.

## 3. Heat Stress and Rapid pH Drop

After slaughter, skeletal muscle undergoes physical structural, and biochemical changes. These changes are triggered by the cession of blood flow and oxygen supply, and the scarcity of glucose resources. Under these conditions, and for postmortem homeostasis purposes, skeletal muscle metabolizes stored glycogen for adenosine triphosphate (ATP) synthesis and use [42]. Lactic acid and hydrogen ions (H^+^) are the endeavor products of several chemical reactions leading to the conversion of glycogen to lactic acid [13]. Since oxygen is lacking, the electron chain is interrupted and pyruvate can no longer enter the mitochondria [42,43]. Hence, lactic acid and H^+^ accumulate, resulting in pH lowering [44]. pH is widely recognized as one of the most accepted indicators of meat quality. Any homeostatic disturbance of postmortem metabolism (e.g., such as rapid pH drop and lower pHu) leads to meat quality defects such as pale, soft, and exudative (PSE) meat, high ultimate pH (pHu) meat (dark, firm, and dry (DFD) meat) and dark-cutting in ruminants [44]. Several studies associated heat stress with a high glycolysis rate and rapid pH decline, resulting in serious damage to skeletal muscle. In broilers under short-term heat stress (36 °C, 1 h), AMP-activated protein kinase (AMPk) activity at 1 h postmortem was greater than that of broilers under thermal neutral (25 °C) conditions [45]. This was also the case with broilers exposed to chronic HS [46]. Moreover, broilers transported during summer (32–42 °C) registered a higher adenosine monophosphate/adenosine triphosphate (AMP/ATP) ratio, increased AMPK, and lower pHu value [47,48]. It is worth noticing that some authors did not register any significant decline in ultimate pH muscle in broilers [15,49].

During and after slaughtering, HS stimulates anaerobic glycolysis within the muscles. The hydrolysis of ATP governed primarily by pyruvate kinase and lactate dehydrogenase in anaerobic conditions then escalates. More pyruvate is converted to lactate leading to an accumulation of H^+^ and lactic acid [50,51]. The result is a rapid pH drop which lowers the water-holding capacity and is at the origin of PSE meat [52,53]. In ruminants, heat stress results in pHu values greater than 5.8, a normal pHu muscle value [18,54]. This finding may be attributed to the effect of the cortisol hormone, which increases under HS conditions. As a result, antemortem skeletal muscle glycogen content is noteworthy reduced and postmortem glycolytic enzyme activity is enhanced [54]. Contrarily, in chickens, heat stress does not affect the antemortem glycogen content [15]. Postmortem glycolytic enzyme activity and pH drops are then faster and greater. Interestingly, some authors demonstrated that high ambient temperature and/or long-term heat exposure may not necessarily have adverse effects on muscle pH and meat quality [47,55]. This could be explained by animals’ adaptation to heat stress [55]. However, rapid pH drops and meat quality damage seem to be associated with short-term exposure to acute ambient temperatures [53,54]. 

## 4. Meat Color and Water Holding Capacity

Besides its detrimental effects on feed intake and growth rate, heat stress was reported to impact physicochemical properties such as color, texture, WHC, and organoleptic properties such as softness, consistency, flavor, and odor in chicken and pork [27,56,57]. Protein denaturation is a result of HS exposure before slaughtering. As proteins are involved in the WHC of meat, each protein damage impedes its ability to bind water. The cumulative effect leads to an impaired WHC marked by high drip and cooking loss [58]. In this trend, numerous studies reported increased values of heat loss and shear force in heat-stressed meat-type broilers [13].

In chickens and pigs, muscles consist of fast-twitching fibers [52,53]. These fibers rely mainly on anaerobic glycolysis [59]. Exposure to stressful ambient temperatures before slaughtering allows for augmented carcass temperature [60], accelerated glycolysis rate, increased H^+^ and lactic acid levels, and elevated protein degradation rate. Consequently, PSE conditions are developed [61]. PSE meat is poorly processed meat, more dry and brittle, and has a poor texture, higher lightness, and lower sensory score [50,52,62] due to its hindered WHC and protein extractability [63,64]. In ruminants, increased pHu values under heat stress conditions negatively affect the shrinkage of the myofilament lattice leading to darker meat color. Many meat quality defects were reported, including higher light absorption, less light scattering, higher oxygen consumption, lower WHC, and increased toughness [18,56,65,66]. However, it seems that these meat quality attributes may be influenced by the duration of heat stress exposure, and the extent of the ambient temperature [20,41,47].

## 5. Impacts of Heat Stress on Muscle Biochemical and Chemical Properties

It is known that ruminants, pigs, and poultry are highly vulnerable to HS due to their rapid metabolism and growth, high production, and species-specific characteristics such as rumen fermentation, transpiration, and skin insulation [67]. Numerous studies have documented how HS affects muscles [68,69]. According to Sula et al. [70], HS results in myocyte fibers that are homogenously eosinophilic, hypereosinophilic, and fragmented. It has been reported that chronic HS would increase the production of lactate in muscle and affect the meat quality. Therefore, acute HS before slaughter accelerates muscle glycogenolysis and increases lactate concentrations in early postmortem slaughter while carcasses are still warm [64]. The result is PSE meat characterized by a decreased WHC, jointly reported in poultry [14,71], but also found recently in cattle [72,73]. Contrarily, animals under chronic HS have diminished muscle glycogen stores, leading to lower lactic acid generation, leading to DFD meat with a higher ultimate pH in ruminants [74], but also in pigs [75]. Additionally, HS effects primarily involve autonomic responses due to the activation of the autonomic nervous system (ANS), which is regulated by catecholamines (adrenaline and norepinephrine) (Table 1). This includes increased respiration and heart rate, increased body temperature, and the redistribution of blood flow from the intestine to the skin for thermoregulation, hence energy utilization from body stores [76] promotes muscle glycogenolysis and inhibits energy storage [74,77]. Both acute and chronic HS cause increases in plasma glucocorticoid concentrations via the activation of the hypothalamic–pituitary–adrenal (HPA) axis. Nevertheless, acute HS leads to increased glucocorticoids more than chronic HS [78]. Glucocorticoids enhance heat loss through vasodilation [79] and increase proteolysis and altered lipid metabolism; proteolysis occurs because of an increased rate of myofibrillar protein degradation in skeletal muscle as mediated by the following mechanisms: the Ca^2+^-dependent-ubiquitin–proteasome, and autophagy–lysosome system [80,81,82] (Table 1).

HS stimulates the hypothalamic–pituitary–adrenal system in poultry, increases the concentration of the circulating hormone corticosterone [83], and has profound effects on protein and lipid metabolism, body composition, and meat quality [84]. Imiku et al. [21] and Lu et al. [85] provided evidence that HS is associated with chemical alterations in chicken meat. High levels of the hormone corticosterone (glucocorticoid) increase fat accumulation in the abdomen, neck, and thighs [86,87,88], but boosted protein degradation and skeletal muscle breakdown [84], potentially via the expression of fatty acid transport protein and the insulin receptor in the pectoralis major [88]. The exposure of animals to HS is related to an elevation in the expression of heat shock proteins in ruminants and pigs [31,89,90], most notably the small alpha βcrystallin (αβC) heat shock protein (sHSP). Heat shock proteins are key components of living muscle that regulate the cytoskeleton and control cell maintenance [91].

In rabbits, HS affects the amount of myoglobin in the muscle, which leads to a decrease in the pigment content of the meat [92].

**Table 1 molecules-28-03332-t001:** Effects of heat stress on biochemical and chemical parameters of the muscle.

Origin	Chemical Class	Sub-Class	References
Pigs	Steroid hormones	Glucocorticoids	[79]
	Carbohydrates	Glycogen	[71]
	Organic acid	Lactic acid	[64]
Ruminants	Protein	Myofibrillar protein	[81]
		alpha βcrystallin (αβC) heat shock protein (sHSP) and HSP27	[31,89,90]
	Steroid hormones	Glucocorticoids	[79]
	Organic acid	Lactic acid	[71]
	Carbohydrates	Glycogen	
	Lipid	Volatile fatty acids	
Broiler	Hormone	Corticosterone	[83]
		Insulin	[88]
	Protein	alpha βcrystallin (αβC) heat shock protein (sHSP) and HSP27	[31,87,89,90]
	Organic acid	Lactic acid	[65]
	Steroid hormones	Glucocorticoids	[79]
	Lipid	Fatty acid	[87]
Rabbit	Organic acid	Lactic acid	[4]
	Lipids	-	[93]
	Proteins	-	[93]
	Metalloprotein	Myoglobin	[92]

## 6. Fatty Acid Profiling

Meat quality is significantly influenced by fat deposition. An animal’s condition or grade of meat may be taken into consideration by a producer when determining the best time to gather the animal. Consumer opinions are even more diverse, with some preferring leaner meats while others prefer fatty meats. These additional requirements from various industry segments make it important to understand why targets exist and how fat accumulates throughout an animal’s life. Schumacher et al. [94] reported that fat deposition occurred after relative muscle growth decreased and continued to increase while bone growth decreased. The rate of adipose tissue growth varies greatly depending on the location and the growth stage. Environmental factors that affect metabolism can affect fat deposition. Chronic HS reduces beta-oxidation and positively affects lipid deposition to reduce thermogenesis [95]. In addition, Heng et al. [96] reported that exposure to HS alters the expression of genes associated with lipid metabolism and storage, leading to increased obesity in piglets born to heat-stressed sows. Kouba et al. [97] concluded that chronically heat-stressed growing pigs showed increased lipid metabolism in both the liver and adipose tissue (lipoprotein lipase activity). This promotes the uptake and storage of plasma triglycerides in adipose tissue, contributing to obesity. In dairy cattle, Hao et al. [98] reported that heat-stressed animals had increased lipogenic capacity but decreased lipolysis. Lu et al. [85] studied the effect of HS on fat deposition in two genetic chickens. Their results showed that the effects of HS were breed-dependent, with increased abdominal and intermuscular fat deposition compared to other types [94]. Lipid oxidation, the process by which meat lipids oxidize and interact with other meat components, causes meat degradation and undesirable nutritional effects [99]. In this context, HS accelerates protein denaturation processes and cell death, leading to the accumulation of ROS and continuous oxidative damage in tissues [100]. Thus, HS damages or destroys mitochondria, causing subsequent changes in energy metabolism pathways. It is now widely accepted that the metabolic changes in energy substance could alter the meat quality attributes in broiler chickens [101].

El-Tarabany et al. [102] showed that HS significantly increased the percentage of abdominal fat in broiler chickens. The prolonged HS significantly increased the contents of saturated fatty acids (SFA) (myristic and palmitic) in the breast and thigh muscles of broiler chickens. Meanwhile, chronic HS decreased the concentrations of monounsaturated fatty acids (MUFA) (myristoleic, palmitoleic, and oleic) in the breast and thigh muscles of the broiler. Moreover, HS significantly decreased the concentration of polyunsaturated fatty acids (PUFA) (linoleic, docosahexaenoic, and eicosapentaenoic) in breast and thigh muscles.

Pig meat was also denatured due to HS. In lean pigs, it has been reported that increased levels of MUFA in *Longissimus* and lower ratio PUFA/SFA in the gluteus after exposure to hot ambient temperature [103,104]. HS has been reported to activate myofascial phospholipase and phosphatidylinositol phosphate kinase for a short time when the ambient temperature rises above 25–35 °C [105]. The impact of short-term HS on lamb showed an increase in muscle omega-6 fatty acid concentration from short-term heat stress may induce oxidative stress via proinflammatory action [6].

## 7. Mineral Composition

Less is known about the impact of animal HS conditions on meat mineral content and its distribution. Under stressful conditions, the acid-base balance of blood is disturbed in heat-stressed animals [106]. The alteration of electrolyte status must be resolved by mineral supplementation. Further, mineral supplementation is necessary because HS can lead to oxidative damage. In general, micronutrients’ ability to enhance performance under heat conditions depends on the species and the physiological stage within each species (growing or breeding animals). In warmer environments, animals consume less food and consequently less mineral intake. In addition, thermoregulatory responses during HS may also influence mineral requirements. The principal pathways of heat loss during heat stress are sweating and panting. Bovines lose a significant amount of minerals via sweat (especially potassium and sodium). As reviewed by Beede and Collier [107] in lactating animals under HS, potassium, and sodium supplementation more than National Research Council (NRC) recommendations resulted in a 3–11% increase in milk production. Electrolyte supplementation (ammonium chloride, potassium chloride, and/or sodium bicarbonate) in potable water or feed decreases the adverse impact of HS in broilers [108]. Nevertheless, the magnitude of the impact of electrolytes relies on the electrolyte balance of the diet [109,110].

In heat-stressed cows, reduced blood bicarbonate concentration because of respiratory alkalosis compromises the buffering capacity of the rumen. Exposure to HS in chickens leads to decreased feed intake, feed efficiency, and weight gain [111,112]. Once domestic poultry (including ducks) are exposed to HS, their hematological index decreases [113]. In addition, it has been reported that HS could generate altered calcium (Ca^2+^) homeostasis in skeletal muscle which could be the result of a disruption of several critical functions [114]. Heat stress induced an elevation in zinc (Zn) levels, which was independent of the restriction of dietary intake. Although the cause of the increased zinc content in heat-stressed pork meat is unknown, it is beneficial to consumers as zinc intake is essential for human health due to its role in cell division and growth, making it a significant nutritional factor in enzyme systems, immune and reproductive function, gene expression, and antioxidant defense [115].

## 8. Antioxidant Status

The body has enzymatic antioxidants (viz., superoxide dismutase, glutathione peroxidase, and catalase) to protect ROS generated due to HS. It is well known that HS is associated with impaired meat quality and the disruption of redox balance [104,116]. In this context, many studies have reported that prolonged HS has a profound effect on muscle metabolism [20,117]. These effects are associated with increased oxidative reactions and the generation of ROS. This disrupts the redox balance that ensures skeletal muscle stability and maintains meat quality [118,119]. Agarwal and Prabhakaran [120] showed that superoxide dismutase (SOD) along with catalase (CAT) and glutathione peroxidase (GPx) scavenges both intracellular and extracellular superoxide radicals and prevents lipid peroxidation (Figure 1).

Lipid oxidation not only negatively influences the sensory characteristics but also the functional characteristics of meat. Studies have shown that high ambient temperatures initiate lipid oxidation in cell membranes. The oxidation of lipids is a significant problem, producing off-flavors and often overcooked flavors in meat [9,122]. The major effects of lipid oxidation on meat quality include changes in odor, color, taste, and texture, adverse effects on functional properties such as protein solubility and WHC, and a decline in the bioavailability of some nutrients [123,124]. A study investigating the effects of HS on lipid oxidation in duck meat showed that HS significantly increased lipoxygenase activity and thiobarbituric acid-reactive substances and decreased the content of free unsaturated fatty acids in duck meat [125]. In addition, a similar report on broilers showed that the lipid oxidation content in meat significantly increased under HS, accompanied by enhanced activities of antioxidant enzymes [126]. Previous studies showed that HS accelerates the oxidation of muscle tissue causing changes in the pro-oxidant/antioxidant balance and compromising meat quality in pigs [104], broilers [127,128], lambs [129,130,131], rabbits [132,133], and beef [134].

### Antioxidant Enzyme Activity 

Antioxidant enzymes play an important role in the detoxification of superoxide radicals, thereby protecting cells from free radical damage [135]. During HS conditions, disturbances in the electron transport chain within the membrane lead to the rapid generation of ROS, disrupting physiological and biochemical mechanisms occurring within the cell [136]. HS shifts the antioxidant-free radical equilibrium towards more free radicals [21]. According to Zhang et al. [20] and Wang et al. [51], HS accelerates the oxidation of muscle tissue. A previous study by Li et al. [137] revealed that the changes in antioxidant enzyme activity are closely associated with high temperature (Table 2). Superoxide dismutase (SOD), catalase (CAT) activity, malondialdehyde (MDA) levels, and total antioxidant capacity (T-AOC) were increased in broiler livers exposed to high temperatures [112,138,139]. A study by Zeng et al. [112] showed alterations in SOD, MDA, CAT, and T-AOC, and an increase in the activities of SOD, CAT, and T-AOC, after HS. Akbarian et al. [140] reported that after acute HS, the activity of antioxidant enzymes (CAT, GSH-Px, and SOD) is greatly increased to protect cells from excessive superoxide formation.

## 9. Conclusions

This review summarizes the effects of HS on the properties and physicochemical components of meat and its antioxidant status. It is well known that HS is inevitable in modern animal husbandry, from farm to slaughterhouse. It is well known that different climate conditions have important effects on the organic and inorganic components of meat. HS potentially affects meat quality via the influence of HS on postmortem muscle pH decline rate and ultimate pH (pHu), water holding capacity, or meat color. In addition, it showed a variable meat quality performance during higher temperatures in different species and breeds of animals. For example, ruminant (lamb, beef) meat had lower brightness and higher terminal muscle pH values during extreme heat, as did dark, firm, and dried meat (DFD). Several new phenomena and mechanisms for changes in meat quality under HS conditions have also been reported, explaining different effects on carcass quality attributes and meat quality attributes, and reducing economic losses for producers and consumers. The effects of heat stress can be mitigated by good farming practices, taking good precautions, and good management. Genetic development and reproductive measures, physical environment modifications, and dietary management are the most important strategies to mitigate heat stress.

## Figures and Tables

**Figure 1 molecules-28-03332-f001:**
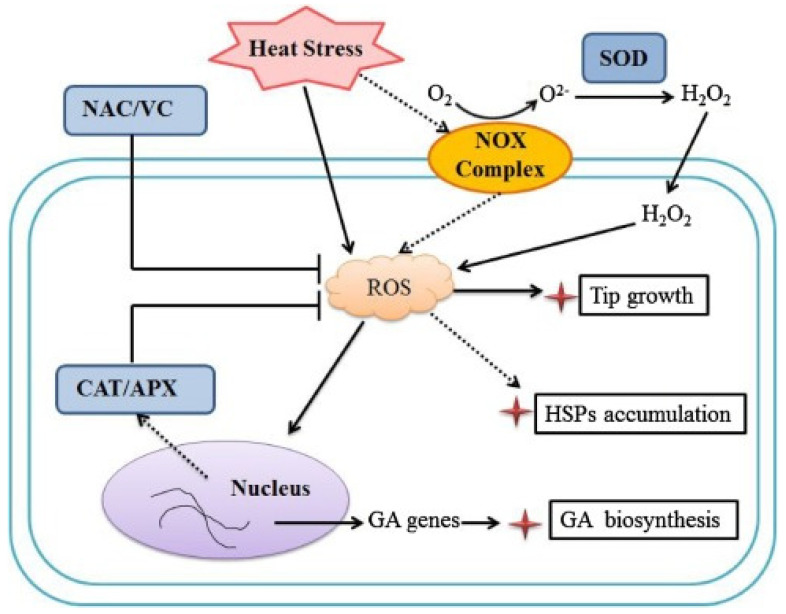
Schematic representation showing the relationship between HS and ROS [121]. GA: ganoderic acid, HSPs: heat shock proteins, ROS: reactive oxygen species, SOD: superoxide dismutase, CAT: catalase, NAC: N-acetyl-l-cysteine, VC: ascorbic acid, APX: Ascorbate Peroxidase.8.1. Lipid Oxidation.

**Table 2 molecules-28-03332-t002:** Variation of antioxidant enzymes activity under heat stress.

Antioxidant Enzyme.	Animal Species	Effects of HS	References
Antioxidant capacity (T-AOC)	Lambs	Increased	[129]
	Broilers	Increased	[128]
	Rabbits	Accelerated	[132]
Superoxide dismutase (SOD)	Pigs	Enhanced	[104]
	Broilers	Increased	[141,142]
	Goats	High	[135]
	Beef	High	[134]
Catalase (CAT)	Pigs	Stimulated	[104]
	Broilers	Increased	[142]
	Lambs	Increased	[129]
Malondialdehybe (MDA)	Pigs	Elevated	[104]
	Broilers	Increased	[138]
Glutathione Peroxidase (GPx)	Pigs	Low activity	[139]
	Broilers	Increased	[142]
	Japanese quail	Decreased	[143]

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
