# Peer review of "Physicochemical Properties, Antioxidant Markers, and Meat Quality as Affected by Heat Stress: A Review"

_molecules, 2023, doi:10.3390/molecules28083332_

Round 1

Reviewer 1 Report

1.      The authors should re-write the abstract starting from the back of the study, objectives and methodology and at the end what will be the benefits of this study for the general community.

2.      Include some data about “assessment of heat stress severity”

3.      Include some data about “heat stress and meat safety”

4.      Include some data about “management strategies for alleviation of heat stress antemortem”.

Author Response

Dear Reviewer 

Thank you for the comments to improve my paper

All comments were taken in account and revised in the manuscript

Cordially

Reviewer 2 Report

I would advise authors to create a list of abbreviations.

When the abbreviation is mentioned for the first time in the main text (excluding the abstract), it is necessary to decode it. (See comments in the attached file.)

Line 66 - Correct registration is required.

Line 126 - What is pHu? Please decode.

Line 127 - What is DFD? Please decode.

Line 130 - What is AMP? Please decode.

Line 130 and Line 133 - Which of these two is correct "AMPk" or "AMPK"?

Line 143 - Which of these two is correct "heat stress" or "heat-stress"? I would recommend using the same terminology and the same notation in the same manuscript.

From line 149 to line 154 the reference is missing [57].

Line 154 - Which of the following is correct "[58;59]" or "[58,59]"?

Line 188 - What is CPM? Please decode.

Line 212 - What is HSP27? Please decode.

The authors have lost chapter 4 in the manuscript.

Line 232 - What is VLDL? Please decode.

From line 234 to line 239 the reference is missing [101].

Line 241 - What is ROS? Please decode.

Line 247 - What is FA? Please decode.

Line 248 - What is MUFA? Please decode.

Line 250 - What is PUFA? Please decode.

Line 254 - What is SFA? Please decode.

Line 273 - What is NRC? Please decode.

Line 285 - What is Ca? Please decode.

Line 288 - What is Fe? Please decode.

Line 289 - What is Zn? Please decode.

Line 303 - What is SOD? Please decode.

From line 341 to line 342, the thought is not understood (the meaning of the punctuation marks is not clear).

I would suggest moving reference [57] to line 505.

I did not find reference [101] in the manuscript.

Author Response

(The authors gave the same response as above.)

Reviewer 3 Report

Influence of heat stress on meat qulaity is reviewed in this manuscript. the subject is useful for meat researcher and industry. However, the manuscript is not well prepared, there are many English writing errors. Moreover, the content is not professional. and there are many errors. for expample, the section 4 is missing. and the content of the manuscript is not well orignaized. Molecules is a high quality jounal, therefore, I suggest reject it. Some questions are marked in the manuscript.

Author Response

(The authors gave the same response as above.)

Reviewer 4 Report

The article entitled "Physicochemical properties, antioxidant markers, and meat quality as affected by heat stress: a review" by Kefi et al., is an interesting review that includes the main effects of heat stress on animals and the meat obtained from them. However, in order to be considered for publication in Foods, the authors must make some improvements that are outlined below.

Line 39: Replace "capacity" with "status".

Line 58: It is the first time that heat stress is mentioned in the main text. Although you have cited the meaning of the acronym in the abstract, following journal guidelines you should cite this initialism the first time it appears in the main text.

Line 127: The meaning of the acronym DFD has not been exposed yet, you should indicate it here.

Line 143: Keep the same format for "ante-mortem" as for "post-mortem" (separator hyphen or not, italics or not). Check the rest of the text accordingly.

Line 144: "Post-mortem" previously it was cited in italics, keep a single way of citing it to be homogeneous throughout the text. Check the rest of the text accordingly.

Lines 154-155: In the previous paragraph you suggested that heat stress did not negatively affect chicken muscle pH values (lines 133-134 and 147-149). However, it is strongly stated here that heat stress has a negative effect on the pH value. Correct this inconsistency.

Line 145: It's a confusing sentence. Replace "Contrary to chickens" with "Contrary, in chickens heat stress…".

Line 154: Add a comma between "color" and "WHC".

Line 154: Replaces ";" by "," in the cited references.

Lines 153-173: Nothing to say about the effect of heat on pigs? As far as I know, PSE meats can become very common in the porcine species, with the consequent economic importance for the meat industry. Could you add information about how heat stress affects the color and water holding capacity of pork meat?

Line 174: Remove the capital letter from "Heat".

Line 187: The meaning of "DFD" should have been explained earlier, so remove it from this part of the text.

Line 188: The meaning of the acronym CPM has not been exposed yet; you should indicate it here.

Line 207: The word "abdomen" is duplicated, delete it once.

Line 216 (Table 1): The table should be completed with how HS affects the chemical and biochemical parameters of the muscle. It is necessary to add an additional column where this information is indicated, since the original table is worthless.

Line 241: The meaning of the acronym "ROS" has not been exposed yet; you should indicate it here.

Line 247: Same as above for "FA". Please check all abbreviations/ acronyms and verify that the first time you name them in the main text you explain their meaning and then do not repeat it again (e.g., SFA, MUFA, PUFA, NRC, etc.).

Line 253: "Longissimus" should appear in italics.

Lines 279-278: This sentence does not refer to HS, why include it here?

Line 306: Include the meaning of the acronyms in the figure in the figure caption.

Line 334: Add a space between "temperatures" and "[115...].

Line 339 (Table 2): I don't think a row for antioxidant capacity should be included here, as this takes into account a multitude of antioxidant compounds (phenolic acids, polyphenols, bioactive peptides, etc.) and not just antioxidant enzymes.

Lines 341-342: Replaces the semicolon and semicolon with commas.

Author Response

(The authors gave the same response as above.)

Round 2

Reviewer 3 Report

The manuscript has been revised properly.